# Toward Optimal ANC: Establishing Mutual Information Lower Bound

## Abstract

Active Noise Cancellation (ANC) algorithms aim to suppress unwanted acoustic disturbances by generating anti-noise signals that destructively interfere with the original noise in real time. Although recent deep learning–based ANC algorithms have set new performance benchmarks, there remains a shortage of theoretical limits to rigorously assess their improvements. To address this, we derive a unified lower bound on cancellation performance composed of two components. The first component is information-theoretic: it links residual error power to the fraction of disturbance entropy captured by the anti-noise signal, thereby quantifying limits imposed by information-processing capacity. The second component is support-based: it measures the irreducible error arising in frequency bands that the cancellation path cannot address, reflecting fundamental physical constraints. By taking the maximum of these two terms, our bound establishes a theoretical ceiling on the Normalized Mean Squared Error (NMSE) attainable by any ANC algorithm. We validate its tightness empirically on the NOISEX dataset under varying reverberation times, demonstrating robustness across diverse acoustic conditions.

## 1 Introduction

Active Noise Cancellation algorithms Lueg (1936); Nelson & Elliott (1991); Fuller et al. (1996); Hansen et al. (1997); Kuo & Morgan (1999) aim to improve listening environments by generating anti-noise signals that destructively interfere with unwanted disturbances in real time. Classical adaptive-filtering methods—most notably the Filtered-x Least Mean Squares (FxLMS) algorithm Boucher et al. (1991) and its variants Das & Panda (2004); Patra et al. (1999); Tan & Jiang (2001); Kuo & Wu (2005); Tobias & Seara (2005), have been used to tackle this problem, but they often struggle in non-stationary or highly reverberant settings and offer limited guidance on how close they operate to fundamental performance limits.

In recent years, deep learning–based ANC models such as DeepANC Zhang & Wang (2021), Attentive Recurrent Networks (ARN) Pandey & Wang (2022), and DeepASC Mishaly et al. (2025) learn end-to-end mappings from reference and error microphones to cancellation waveforms, outperforming classical signal processing methods. Variants employing recurrent CNNs Park et al. (2023); Mostafavi & Cha (2023), autoencoders Singh et al. (2024), and fully connected nets Pike & Cheer (2023), together with selective fixed-filter schemes Shi et al. (2020); Park & Park (2023), Kalman-filter hybrids Luo et al. (2023c), and attention modules Zhang et al. (2023a), further enhance robustness and low-latency control. However, no unified theory specifies the minimal achievable error under given acoustic and algorithmic constraints, complicating optimality assessment and targeted architecture design. To fill this gap, we derive a unified lower bound on ANC performance that blends two complementary perspectives:

- An information-theoretic term that links the residual error power to the fraction of disturbance entropy captured by the anti-noise signal—quantifying limits imposed by information-processing capacity.

- A support-based term that measures the irreducible error arising from frequency bands unsupported by the cancellation path—reflecting fundamental physical constraints.

By taking the maximum of these two terms, our bound establishes the first general theoretical ceiling on the NMSE attainable by any ANC algorithm, regardless of model complexity. We stress that this

lower bound framework adequate to every possible ANC algorithm, may it be a neural network trained on dataset, a linear filtering, or adaptive mechanism.

We validate our contributions through extensive experiments on the NOISEX benchmark under varying reverberation times. The primary contributions of this work are summarized as follows:

- **Theoretical benchmark:** We derive a unified lower bound on ANC performance that accounts for both information-processing and spectral-support constraints.
- **Empirical validation:** We confirm the tightness of our bound over various noise types and for different reverberation times. Boundary implementations are released in the supplementary material.

## 2 RELATED WORK

The foundational principle of destructive interference was introduced by Lueg, who demonstrated that appropriately phased signals can cancel unwanted oscillations (Lueg, 1936). Early adaptive schemes adopted the Least Mean Squares (LMS) criterion to track variations in amplitude and phase, achieving robust echo cancellation in acoustic feedback paths Burgess (1981); Nelson & Elliott (1991); Fuller et al. (1996); Hansen et al. (1997); Kuo & Morgan (1999). To account for the dynamics of the primary and secondary acoustic paths, the FxLMS algorithm was proposed, with subsequent analysis quantifying the impact of secondary-path inversion errors on convergence and stability Boucher et al. (1991).

Extensions of the LMS framework have focused on mitigating real-world nonlinearities. Functional-Link Artificial Neural Networks (FLANN) were integrated into LMS filters to compensate for distortion in the control path Patra et al. (1999). Bilinear filtering improved modeling of mild nonlinearities Kuo & Wu (2005). A tangential-hyperbolic adaptation further captured loudspeaker saturation effects, ensuring stable cancellation under high-amplitude conditions Ghasemi et al. (2016).

Data-driven models have more recently transformed ANC by learning end-to-end mappings from sensor measurements to cancellation signals. Convolutional-LSTM networks estimate both amplitude and phase responses Zhang & Wang (2021), and recurrent convolutional architectures exploit temporal dependencies for improved active cancellation prediction Park et al. (2023); Mostafavi & Cha (2023); Cha et al. (2023). Fully connected network also applied to ANC Pike & Cheer (2023) as well as Autoencoder-based encodings scheme have been employed to extract latent features and generalize across diverse noise conditions Singh et al. (2024). Selective Fixed-filter ANC (SFANC) leverages pre-trained filter banks chosen by lightweight neural controllers, enabling rapid adaptation without full retraining Shi et al. (2020; 2022; 2023a); Luo et al. (2022); Park & Park (2023).

The works of Luo et al. (2023b;a; 2024) have focused on synthesizing optimized filter banks for selective fixed-filter ANC. Multichannel ANC systems exploit spatial diversity to achieve superior noise attenuation. Deep multichannel controllers learn inter-channel relationships for ANC Zhu et al. (2021); Zhang & Wang (2023); Antoñanzas et al. (2023); Xiao et al. (2023); Zhang et al. (2023b); Shi et al. (2023b; 2024). Zhang et al. (2023a) proposed an attention-driven framework for real-time ANC by integrating the Attentive Recurrent Network of Pandey & Wang (2022), and Zhang et al. (2022) demonstrated real-time ANC using attentive recurrent architectures for online adaptation. In addition, metaheuristic strategies have been explored, Zhou et al. (2023) applied a genetic algorithm to optimize ANC filters, and Ren & Zhang (2022) employed a bee colony optimization scheme to refine control coefficients.

## 3 BACKGROUND - TYPICAL ANC SETUP

As we can observe in Figure 1, a typical ANC feedforward setup, the signal captured by the reference microphone is denoted as $x(n)$ and is transmitted through the primary path $P(z)$, and their convolution produces the primary signal denoted $d(n)$ and defined as $d(n) = P(z) * x(n)$. The signal $x(n)$ and the signal captured by the error microphone $e(n)$ are used as inputs of the ANC system to generate the cancelation signal $y(n)$, which is played by the loudspeaker $f_{LS}$. The produced signal $f_{LS}\{y(n)\}$, supposed to eliminate the undesirable noise around the error microphone, is then transmitted through the secondary path $S(z)$ to generate the anti-signal $a(n)$ defined by,

$a(n) = S(z) * f_{LS}\{y(n)\}$ Then, the error signal $e(n)$ is obtained by calculating the difference between the primary and secondary path's outputs, $e(n) = d(n) - a(n)$. The ANC controller aims to reduce the error signal $e(n)$ to zero, achieving optimal noise cancellation. In the feedback ANC method, only $e(n)$ is used to generate the canceling signal, focusing on minimizing the residual noise captured by the error microphone.

## 4 FUNDAMENTAL LIMITS ON CANCELLATION PERFORMANCE: INFORMATION-THEORETIC BOUNDS

We address the problem of canceling an undesired disturbance signal, denoted as a discrete-time random process $d(n)$, which arises from filtering a reference source signal $x(n)$ through a system characterized by the primary path transfer function $P(z)$. This scenario is canonical in applications such as ANC, acoustic echo cancellation, and adaptive interference mitigation. The objective is to synthesize a cancellation signal $y(n)$ using an adaptive algorithm or model, potentially involving non-linear transformations $f(\cdot)$ (e.g., modeling secondary path dynamics or actuator non-linearities), such that the residual error signal $e(n) = d(n) - f(y(n))$ is minimized in a suitable sense, typically minimizing its NMSE or power.

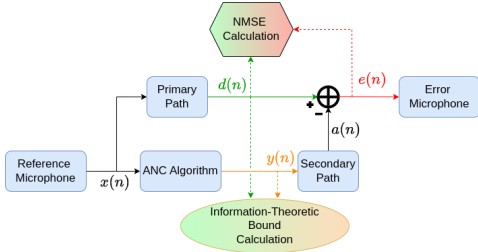

Figure 1: High-level schematic of a feedforward ANC system with lower-bound calculation.

To establish fundamental benchmarks against which practical algorithms can be evaluated, we derive a lower bound on the achievable cancellation performance rooted in information theory. This bound quantifies the irreducible error stemming from the information processing limitations inherent in generating $y(n)$ based on available observations.

We posit that the reference signal $x(n)$ is a zero-mean Wide-Sense Stationary (WSS) random process. Consequently, if the primary path $P(z)$ represents a linear time-invariant (LTI) system, the disturbance signal $d(n) = P(z)x(n)$ is also a zero-mean WSS process. Furthermore, we assume the synthesized signal $y(n)$ and the disturbance $d(n)$ are jointly WSS and ergodic. Let $S_{dd}(e^{j\omega})$ denote the power spectral density (PSD) of $d(n)$. The average power of the disturbance is given by its variance $\sigma_d^2 = \mathbb{E}[|d(n)|^2]$. A cornerstone connecting the time-domain correlation structure of stationary signals to their frequency-domain power distribution is the Wiener-Khinchin Theorem.

[Wiener-Khinchin Theorem for Discrete-Time WSS Processes] Let $v(n)$ be a discrete-time WSS random process with Auto-Correlation Function (ACF) defined as $R_{vv}(\tau) = \mathbb{E}[v(n)v^*(n-\tau)]$. Its Power Spectral Density (PSD), $S_{vv}(e^{j\omega})$, is the Discrete-Time Fourier Transform (DTFT) of its ACF:

$$S_{vv}(e^{j\omega}) = \sum_{\tau=-\infty}^{\infty} R_{vv}(\tau)e^{-j\omega\tau}$$

Conversely, the ACF can be recovered from the PSD via the inverse DTFT:

$$R_{vv}(\tau) = \frac{1}{2\pi} \int_{-\pi}^{\pi} S_{vv}(e^{j\omega})e^{j\omega\tau}d\omega$$

The average power of the process is the integral of its PSD over the fundamental frequency interval (or equivalently, the ACF at lag zero):

$$\mathbb{E}[|v(n)|^2] = R_{vv}(0) = \frac{1}{2\pi} \int_{-\pi}^{\pi} S_{vv}(e^{j\omega})d\omega$$

This theorem justifies the use of spectral representations for power calculations under the WSS assumption. For our disturbance signal $d(n) = P(z)x(n)$, its PSD is related to the PSD of the source $S_{xx}(e^{j\omega})$ by $S_{dd}(e^{j\omega}) = |P(e^{j\omega})|^2 S_{xx}(e^{j\omega})$, assuming $P(z)$ is stable. The average power of the disturbance is thus:

$$\sigma_d^2 = \mathbb{E}[|d(n)|^2] = \frac{1}{2\pi} \int_{-\pi}^{\pi} S_{dd}(e^{j\omega})\,d\omega = \frac{1}{2\pi} \int_{-\pi}^{\pi} |P(e^{j\omega})|^2 S_{xx}(e^{j\omega})\,d\omega \tag{1}$$

Now, consider the information-theoretic perspective. Let $H(d)$ denote the differential entropy rate of the disturbance process $d(n)$, assuming it exists and is finite. Let $I(y; d)$ denote the mutual information rate between the synthesized signal process $y(n)$ and the disturbance process $d(n)$. Mutual Information measures the information that $d(n)$ and $y(n)$ share and therefore quantifies how much knowing one of these variables reduces uncertainty about the other. Intuitively, effective cancellation requires $y(n)$ (or more precisely, $f(y(n))$), where $f$ is some processing function, to be highly informative about $d(n)$. Building upon rate-distortion theory concepts, a lower bound on the residual error power $\sigma_e^2 = \mathbb{E}[|e(n)|^2]$ can be established:

[Information-Theoretic Bound on Cancellation Error] Under the assumption that $y(n)$ and $d(n)$ are jointly WSS and ergodic, and that $d(n)$ possesses a finite differential entropy rate $H(d)$, the average power of the residual error $e(n) = d(n) - f(y(n))$ is lower bounded by:

$$\sigma_e^2 = \mathbb{E}[|e(n)|^2] \geq \sigma_d^2 \left(1 - \frac{I(y(n); d(n))}{H(d(n))}\right) \tag{2}$$

where $\sigma_d^2 = \mathbb{E}[|d(n)|^2]$ is the average power of the disturbance signal.

Let $d(n)$ and $y(n)$ be jointly wide-sense stationary (WSS) and ergodic random processes. Suppose $d(n)$ has finite variance $\sigma_d^2$ and finite differential entropy rate $H(d)$. Define the residual error as:

$$e(n) = d(n) - f(y(n))$$

where $f : \mathbb{R} \to \mathbb{R}$ is any deterministic mapping (possibly nonlinear). Let $\hat{d}(n) = f(y(n))$ be the estimate of $d(n)$ based on $y(n)$.

From information theory, the mutual information between $d(n)$ and $y(n)$ satisfies:

$$I(d(n); y(n)) \geq R(D) \tag{3}$$

where $R(D)$ is the rate-distortion function of the source $d(n)$ at distortion level $D = \mathbb{E}[(d(n) - \hat{d}(n))^2] = \sigma_e^2$.

For general stationary ergodic sources with finite differential entropy rate $H(d)$, the Shannon lower bound on the rate-distortion function provides:

$$R(D) \geq H(d) - \frac{1}{2}\log(2\pi e D) \tag{4}$$

Solving equation 4 for $D$ yields:

$$D \geq \frac{1}{2\pi e} \cdot \exp\left(2(H(d) - I(d(n); y(n)))\right) \tag{5}$$

However, this expression is difficult to interpret directly. Instead, we derive a more interpretable but looser bound by noting that mutual information is always upper bounded by entropy:

$$I(d(n); y(n)) \leq H(d)$$

Hence, we can define the normalized mutual information ratio:

$$\alpha := \frac{I(d(n); y(n))}{H(d)} \in [0, 1]$$

Recall, that for gaussian noise, the entropy is maximazied and the following holds:

$$\sigma_d^2 = \frac{1}{2\pi e} e^{2H(d)}$$

Rearranging terms, of equation 5, and keep in mind that the ratio is always smaller than one, so for negative exponent it is upper bound,

$$D \geq \frac{1}{2\pi e} \cdot \exp\left(2H(d)\right) \cdot \exp\left(-\frac{I(d(n); y(n))}{H(d)}\right)$$

expanding the second exponent as taylor series, and taking the first order yeilds,

$$\sigma_e^2 \geq \sigma_d^2 \left(1 - \frac{I(d(n); y(n))}{H(d)}\right) \tag{6}$$

Intuitively, this ratio measures the fraction of the information in $d(n)$ that is captured by $y(n)$. If $\alpha = 1$, perfect reconstruction is possible in principle, and $\sigma_e^2 \to 0$. If $\alpha = 0$, then $y(n)$ carries no information about $d(n)$, and $\sigma_e^2 \geq \sigma_d^2$.

We thus propose a linear lower bound on the distortion based on this ratio:

$$\sigma_e^2 \geq \sigma_d^2(1 - \alpha) = \sigma_d^2 \left( 1 - \frac{I(d(n); y(n))}{H(d)} \right) \tag{7}$$

which completes the proof.

This fundamental bound highlights that the efficacy of any cancellation strategy is limited by how much information about the target disturbance $d(n)$ is effectively encoded in the synthesized signal $y(n)$. If $y(n)$ carries no information about $d(n)$ ($I(y; d) = 0$), the bound implies $\sigma_e^2 \geq \sigma_d^2$, meaning no cancellation is achieved on average. Conversely, if $y(n)$ allows for perfect reconstruction such that $f(y(n)) = d(n)$, then $I(y; d)$ must approach $H(d)$ (under suitable conditions, e.g., Gaussianity), and the bound approaches zero. However, achieving $I(y; d) = H(d)$ may be impossible in practice due to several factors not explicitly captured in the bound's derivation but reflected in the properties of $y(n)$ generated by real-world algorithms. These include:

- **Causality:** Practical algorithms operate causally, potentially requiring a significant observation window (context) to accurately estimate latent variables (e.g., channel coefficients, internal states) needed to generate an effective $y(n)$. During initialization or adaptation phases, $I(y; d)$ might be substantially lower than its asymptotic potential.

- **Model Mismatch:** The assumed structure relating $x(n)$ to $d(n)$ (given by $P(z)$) or the model used to generate $y(n)$ (including $f(\cdot)$) might not perfectly match reality.

- **Computational Constraints:** The complexity of the algorithm generating $y(n)$ might limit its ability to extract and encode all relevant information about $d(n)$.

Therefore, the bound equation 2 represents an idealized limit conditioned on the statistical properties (specifically, the joint distribution encoded in $I(y; d)$) of the output $y(n)$ produced by a given cancellation system. We can express this bound in terms of the NMSE, often reported in decibels (dB), defined as $\mathrm{NMSE} = \mathbb{E}[|e(n)|^2]/\mathbb{E}[|d(n)|^2]$. Substituting equation 1 and converting to dB:

$$\mathrm{NMSE_{dB}} = 10 \log_{10} \left( \frac{\mathbb{E}[|e(n)|^2]}{\mathbb{E}[|d(n)|^2]} \right) \geq 10 \log_{10} \left( 1 - \frac{I(y; d)}{H(d)} \right) + 10 \log_{10}(\|P(z)\|_2^2) \tag{8}$$

This information-theoretic bound is inherently dependent on the specific algorithm (as it determines the statistics of $y(n)$ and thus $I(y; d)$). It serves as a valuable theoretical ceiling, quantifying performance limitations arising purely from the representational capacity and information-processing capabilities of the cancellation architecture.

## 5 System Support Constraints and Model-Independent Limits

Beyond information-theoretic limitations, cancellation performance can be constrained by fundamental physical or structural properties of the system, irrespective of the algorithm's sophistication. A critical constraint arises from the frequency-domain support of the system components, particularly when the cancellation mechanism $f(y(n))$ cannot generate signals covering the entire frequency spectrum of the disturbance $d(n)$.

Let us specialize to the common case where the cancellation mechanism is an LTI system, $f(y(n)) = S(z)y(n)$, representing the secondary path in ANC or the echo path model in echo cancellation. Let $P(e^{j\omega})$ and $S(e^{j\omega})$ be the frequency responses of the primary and secondary paths, respectively. We define the frequency support of a system as the set of frequencies where its response is non-zero. Let

$$\mathrm{supp}(P) = \{\omega \in [-\pi, \pi] \mid P(e^{j\omega}) \neq 0\}$$

$$\mathrm{supp}(S) = \{\omega \in [-\pi, \pi] \mid S(e^{j\omega}) \neq 0\}$$

If there exists a frequency band where the primary path transmits energy (i.e., $\omega \in \mathrm{supp}(P)$) but the secondary path has no response (i.e., $\omega \notin \mathrm{supp}(S)$), then the component of the disturbance $d(n)$ at

frequency $\omega$ cannot be cancelled by any signal $y(n)$ filtered through $S(z)$. The cancellation signal $f(y(n)) = S(z)y(n)$ will have zero energy at such frequencies.

Let $\Omega_{uncancelable} = \text{supp}(P) \setminus \text{supp}(S)$ denote the set of frequencies where the disturbance exists but the cancellation path has no gain. The portion of the disturbance power residing in this uncancelable frequency region constitutes an irreducible component of the error power. The power of $d(n)$ restricted to this frequency set is given by:

$$\sigma_{d,\text{uncancelable}}^2 = \frac{1}{2\pi} \int_{\Omega_{uncancelable}} S_{dd}(e^{j\omega}) \, d\omega = \frac{1}{2\pi} \int_{\Omega_{uncancelable}} |P(e^{j\omega})|^2 S_{xx}(e^{j\omega}) \, d\omega \quad (9)$$

Since the cancellation signal $f(y(n))$ has zero energy in $\Omega_{uncancelable}$, the residual error $e(n) = d(n) - f(y(n))$ must contain at least this amount of power. This yields a model-independent lower bound on the error power:

[Support-Based Bound on Cancellation Error] Assuming $d(n) = P(z)x(n)$ and $f(y(n)) = S(z)y(n)$, where $x(n)$ is WSS and $P(z), S(z)$ are LTI systems, the average power of the residual error $e(n)$ is lower bounded by the power of the disturbance in the frequency region where the primary path has support but the secondary path does not:

$$\sigma_e^2 = \mathbb{E}[|e(n)|^2] \geq \sigma_{d,\text{uncancelable}}^2 = \frac{1}{2\pi} \int_{\text{supp}(P) \setminus \text{supp}(S)} S_{dd}(e^{j\omega}) \, d\omega \quad (10)$$

This bound holds regardless of the algorithm used to generate $y(n)$, even if it possessed infinite computational power and perfect knowledge of the system, as it stems from the inherent inability of the system $S(z)$ to counteract certain frequency components of $d(n)$. The corresponding NMSE lower bound is obtained by normalizing this irreducible error power by the total disturbance power $\sigma_d^2$:

$$\text{NMSE} = \frac{\mathbb{E}[|e(n)|^2]}{\sigma_d^2} \geq \frac{\sigma_{d,\text{uncancelable}}^2}{\sigma_d^2} = \frac{\int_{\text{supp}(P) \setminus \text{supp}(S)} S_{dd}(e^{j\omega}) \, d\omega}{\int_{-\pi}^{\pi} S_{dd}(e^{j\omega}) \, d\omega} \quad (11)$$

In decibels:

$$\text{NMSE}_{\text{dB}} \geq 10 \log_{10} \left( \frac{\int_{\text{supp}(P) \setminus \text{supp}(S)} |P(e^{j\omega})|^2 S_{xx}(e^{j\omega}) \, d\omega}{\int_{-\pi}^{\pi} |P(e^{j\omega})|^2 S_{xx}(e^{j\omega}) \, d\omega} \right) \quad (12)$$

This expression quantifies the fraction of the total disturbance power that lies in the frequency bands structurally unreachable by the cancellation path $S(z)$. It represents a fundamental performance ceiling imposed by the physical setup or inherent characteristics of the primary and secondary paths.

## 6  UNIFIED PERFORMANCE BOUND

Combining the insights from the information-theoretic and support-based analyses, we arrive at a unified lower bound on the achievable cancellation performance. The actual residual error power must satisfy both constraints simultaneously. Therefore, the NMSE is lower bounded by the maximum of the two individual bounds:

[Unified Lower Bound on NMSE] Under the assumptions of joint WSS for $d(n), y(n)$, finite entropy rate $H(d)$, LTI primary path $P(z)$, and LTI cancellation path $S(z)$, the Normalized Mean Squared Error (NMSE) in decibels is lower bounded by:

$$\text{NMSE}_{\text{dB}} \geq \max \left\{ 10 \log_{10} \left( 1 - \frac{I(y;d)}{H(d)} \right), 10 \log_{10} \left( \frac{\int_{\text{supp}(P) \setminus \text{supp}(S)} S_{dd}(e^{j\omega}) \, d\omega}{\int_{-\pi}^{\pi} S_{dd}(e^{j\omega}) \, d\omega} \right) \right\} \quad (13)$$

This unified bound encapsulates the dual nature of performance limitations in cancellation systems. Performance can be bottlenecked either by the algorithm's inability to capture and utilize sufficient information about the disturbance (the $I(y;d)/H(d)$ term) or by the physical inability of the cancellation path to address certain frequency components of the disturbance (the spectral support mismatch term). This provides a comprehensive theoretical tool for evaluating the fundamental limits of cancellation systems, guiding algorithm design and system configuration. Understanding which bound is dominant in a particular scenario can inform whether efforts should focus on improving the information processing capabilities of the algorithm or on modifying the physical system (e.g., actuator placement, speaker characteristics) to improve spectral coverage.

## 7 EXPERIMENTS

**Evaluation data:** We report performances on the customary NOISEX dataset Varga & Steeneken (1993), which includes a collection of real-world noise recordings commonly encountered in speech and audio processing tasks, including babble, factory, and engine noises. To align with our prepro­cessing pipeline, all samples were standardized to 3 seconds and resampled at 16 kHz. **Simulator:** A rectangular room of dimensions [3, 4, 2] meters (width, length, height) was simulated. Room impulse responses (RIRs) were generated (Allen & Berkley, 1979) via the Python *rir_generator* package, with a 512-tap length and high-pass filtering enabled. Microphone and speaker positions were set at [1.5, 3, 1] m (error mic), [1.5, 1, 1] m (reference mic), and [1.5, 2.5, 1] m (cancella­tion speaker). Customary reverberation times were sampled from {0.15, 0.175, 0.2, 0.225, 0.25} s in the boundary calculations. Although the Scaled Error Function (SEF) Tobias & Seara (2006) is commonly used to model loudspeaker nonlinearity, in this work we consider only the linear case, corresponding to $\eta^2 \to \infty$. **Hyperparameters:** In order to reproduce the performances presented in DeepASC Mishaly et al. (2025) we reproduced the training based on their hyperparameters de­scription. **Baseline Methods:** We confronted our bound derived above with the state-of-the-art deep learning-based ANC methods named DeepANC Zhang & Wang (2021), Attentive Recurrent Network (ARN) Zhang & Wang (2023) and, DeepASC Mishaly et al. (2025). To evaluate the effectiveness of each method under varying noise conditions, we employed pretrained models as a standardized benchmark. This approach ensured consistent performance assessment across different noise types. **Ressources:** All experiments were conducted using NVIDIA RTX 3090 GPUs.

## 8 RESULTS AND DISCUSSIONS

### 8.1 BOUNDARIES HOLD NMSE PERFORMANCES OVER VARIOUS REVERBERATION TIMES

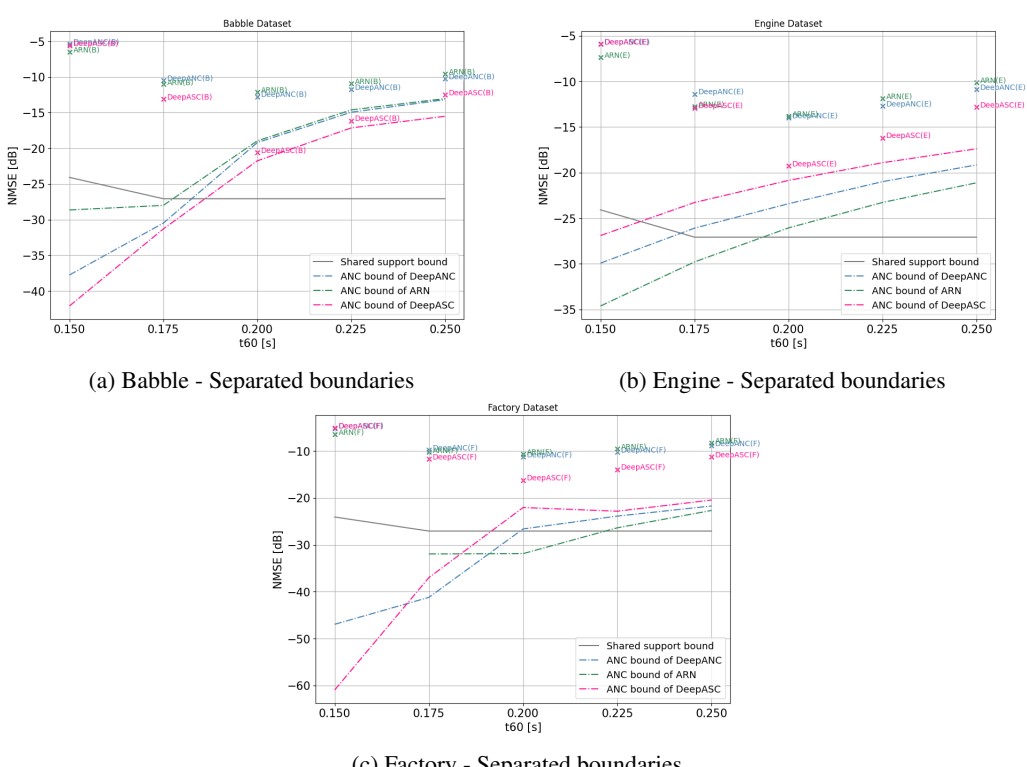

(a) Babble - Separated boundaries  (b) Engine - Separated boundaries

(c) Factory - Separated boundaries

Figure 2: Model performances for various reverberation times with associated boundaries

Figure 2 presents the performances of three baseline methods on three noise types from the NoiseX dataset - Babble, Engine, and Factory noises - and the corresponding boundaries derived in the previous sections for various reverberation times sampled from {0.15, 0.175, 0.2, 0.225, 0.25}. We

show the information-theoretic bound separated from the bound based on support constraints from the ANC method in order to highlight the impact of each term of the unified boundary Equation 13.

**Separated Boundaries**  Firstly, we notice that the bound based on support constraints (Eq. 12) is model and dataset independent. Indeed, the subfigures from Figure 2 share the same support-based boundary (in gray). This bound holds for all the reported performances obtained from the three different methods. We also observe that the Information-Theoretic bound (ANC bound - Eq. 8) is increasing with reverberation times. As reverberation time increases, the signal becomes more complex since the physical translation is the creation of more bounces on the simulator's virtual walls. This leads to longer, more intricate room impulse responses, making both the primary and secondary paths temporally extended and harder to model accurately. This complexity is further amplified by the dense overlap of reflected waves, which blurs the distinction between direct and reflected components, making path estimation more challenging. Consequently, this growing complexity contributes to the steady increase in the ANC information-theoretic boundaries.

**Audio Noise Cancellation Performances**  Figure 2 shows the performances of three models (Deep-ASC, DeepANC and ARN) measured using NMSE over three different noise types. Contrary to what is often observed, the performances are not only calculated and reported for one reverberation time (usually $t_{60} = 0.2s$ Zhang & Wang (2021; 2023); Mishaly et al. (2025)) but for all the reverberation times used in the training process ($t_{60} \in \{0.15, 0.175, 0.2, 0.225, 0.25\}$). In Figure 2 each subfigure corresponds to a different type of noise (respectively Babble, Engine, and Factory noises). From left to right, we observe lower NMSE performances than the previous one. Based on these observations, Babble noise appears to be the easiest to cancel, followed by Engine noise, while Factory noise proves to be the most challenging. Figure 3 provides visual confirmation of these observations by displaying the spectrograms of the noisy signals. The Engine noise exhibits temporally repetitive frequency structures, which enhance its predictability and facilitate its attenuation using adaptive filtering methods. In contrast, the Factory noise shows a more broadband and statistically uniform distribution, lacking distinct temporal patterns, which increases its resistance to cancellation. Furthermore, Engine contains more high-frequency energy than Babble, which may reduce the effectiveness of noise suppression techniques due to the higher spectral variability and lower correlation with past signal frames.

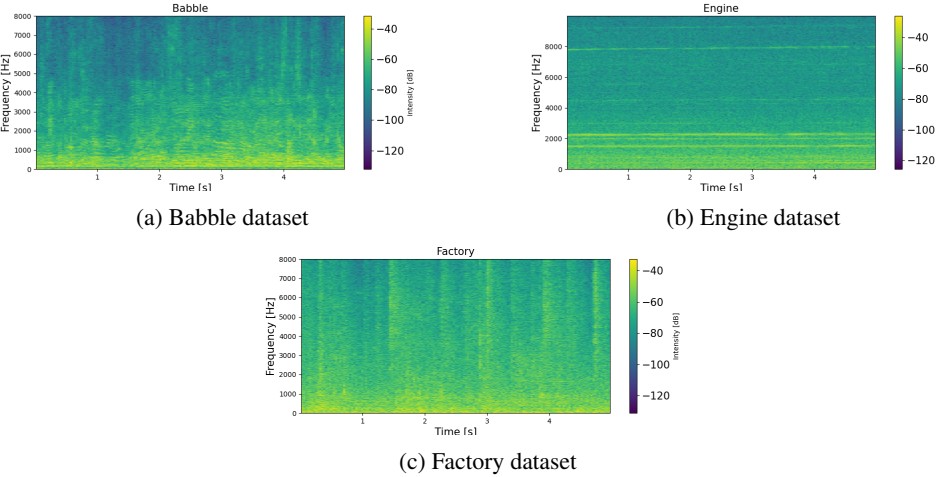

(a) Babble dataset
(b) Engine dataset

(c) Factory dataset

Figure 3: Time-Frequency analysis of the different noise

**Unified Boundaries**  By applying the max operator between the support bound and the ANC bound in Figure 2 the unified boundaries are obtained . It demonstrates that the unified bound derived from Equation 13 is consistently satisfied across the three methods and types of noise. According to previous analysis, we observe that the unified bound necessitates the two terms because the support bound leads for low reverberation times and the Information-Theoretic bound for high reverberation times. Another observation is the increasing of the gap between the performances and the boundaries as noise complexity increases (Figure 2). As previously discussed and illustrated in Figure 3, these observations indicate an increasing level of cancellation difficulty: Babble, followed by Engine,

and finally Factory, which presents the greatest challenge—reflected in a larger gap between actual performance and theoretical boundaries.

Therefore the gap between the performance and the boundary changes through noise types. Also, Figure 4 in Appendix A.3 is a more general observation of the performances that can be used in various fields. Indeed, performance values and deviations from a theoretical boundary can lead to a better understanding of data or algorithms. This work proposes theoretically derived performance boundaries that offer a principled benchmark for analyzing and comparing active noise control algorithms. By unifying information-theoretic and support-based criteria, the framework provides insight into the fundamental limitations of system performance under varying noise conditions, and can inform the design of more effective adaptive filtering strategies in the future.

## 9 LIMITATIONS AND BROADER IMPACT

**Limitations:** While deriving a lower bound for various ANC methods provides valuable insight into their capabilities, such bound estimations come with inherent limitations. In particular, computing the Information-Theoretic bound necessitates a large number of samples to accurately estimate the underlying probability density functions of the signals. Insufficient sampling leads to poor approximation of the true distributions, which in turn distorts the bound calculations, making them unreliable or overly optimistic. Consequently, to ensure meaningful and accurate Information-Theoretic evaluations in ANC scenarios, a substantial volume of diverse audio data is essential. **Broader Impact:** This work helps identify performance limitations and guides algorithm design. Improved ANC technology can positively impact society by enhancing comfort and health in noisy environments, with applications in consumer electronics, healthcare, and public infrastructure. While generally positive, potential concerns include over-reliance on ANC instead of addressing noise sources directly, and the need for careful integration to ensure fairness and transparency.

## 10 CONCLUSIONS

We presented a unified bound that captures the two fundamental limitations in cancellation systems: the algorithm's capacity to extract information about the disturbance $(I(y; d)/H(d))$ and the physical system's ability to counter specific frequency components. It serves as a theoretical tool for assessing ANC limits and guiding design decisions—helping determine whether improvements should target algorithmic information processing or physical system setup. The unified bound was evaluated across various noise types using baseline methods and consistently provided a lower bound on their observed performance. Analysis of the bound provides valuable insights into the characteristics of different noise types. Specifically, we observe that the bound increases with the complexity of the noise, indicating greater theoretical difficulty in achieving effective cancellation. Additionally, the widening gap between the model performance and the bound for more challenging noises suggests limitations in the models' ability to fully exploit the available information, highlighting areas for potential algorithmic improvement.

## 11 REPRODUCIBILITY STATEMENT

In this paper we introduce a new boundary that aims to help all stakeholders in the field improve their understanding of ANC algorithms. This understanding could lead to more efficient algorithms. To do so, We have made efforts to make our boundary reproducible and easily applicable to algorithms developed by future readers. We provide the code used to calculate the various terms of the boundary and an explanation of the methods used during implementation in the appendix.

## 12 ETHICAL STATEMENT

This work helps identify performance limitations and guides algorithm design. Improved ANC technology can positively impact society by enhancing comfort and health in noisy environments, with applications in consumer electronics, healthcare, and public infrastructure. While generally positive, potential concerns include over-reliance on ANC instead of addressing noise sources directly, and the need for careful integration to ensure fairness and transparency.

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

# A APPENDIX

## A.1 NUMERICAL CALCULATION OF THE INFORMATION-THEORETIC BOUNDARY (EQ. 8)

The numerical estimation of Equation 8 requires the estimation of the marginal and the joint entropies of the signals.

To do so, we calculated the marginal Probability Density Functions (PDFs) associated to the primary signal $d(n)$ and the output of the ANC algorithm $y(n)$. To estimate the marginal and joint PDFs of two time series variables, we developed a kernel-based approach that aggregates density estimates across paired sequences. For each pair of subsequences in the datasets, we first apply Gaussian Kernel Density Estimation (KDE) to compute the marginal PDFs and joint PDF over fixed evaluation grids. Bandwidth is set inversely proportional to the number of histogram bins (bin_count) to control the smoothness of the estimates. The joint and marginal PDFs are normalized using numerical integration to ensure valid probability distributions. These estimates are then averaged across all subsequence pairs to yield the final marginal and joint PDFs.

We compute an information-theoretic bound by estimating the mutual information (MI) between predicted and observed sequences using entropy terms derived from their joint and marginal probability distributions. Entropies are calculated via normalized histograms obtained from KDE-based PDF estimates. The MI is then used to derive a coefficient that scales the energy of the direct path in the room impulse response, yielding an upper-bound estimate and its decibel-scaled version.

## A.2 NUMERICAL CALCULATION OF THE SYSTEM SUPPORT-BASED BOUNDARY (EQUATION 12)

To assess the spectral separation between primary and secondary acoustic paths, we compute a support-based boundary metric in the frequency domain. The method performs a Fast Fourier Transform (FFT) on the impulse responses of both paths and identifies the spectral support of the primary path as the set of frequency bins that exceed a predefined magnitude threshold (e.g. 45 dB). A boundary score is then calculated as the proportion of these primary-support bins that do not overlap with the secondary path's support (defined as bins below the threshold). This ratio quantifies the distinctness of the primary path and is reported both linearly and in decibels.

## A.3 BOUNDARY OVERVIEW FOR A FIXED REVERBERATION TIME ($t_{60} = 0.2s$)

In this section, we provide a consolidated overview of the boundaries across different noise types to follow standard machine learning conventions and offer a comprehensive view of the results. Indeed, Figure 4 shows the performances and the discussed boundaries but this time for $t_{60} = 0.2s$. As previously mentioned, the unified boundary—defined as the maximum of the Information-Theoretic and Support-based boundaries—encloses the performance achieved by the different methods across all noise types. Figure 4 highlights the decline in performance and the increasing gap between performances and theoretical boundaries as the noises become more complex (from left to right along the x-axis in Figure 4). A key point is that, for a given algorithm, the lower bound depends on the noise characteristics as it changes the equivalent channel capacity.

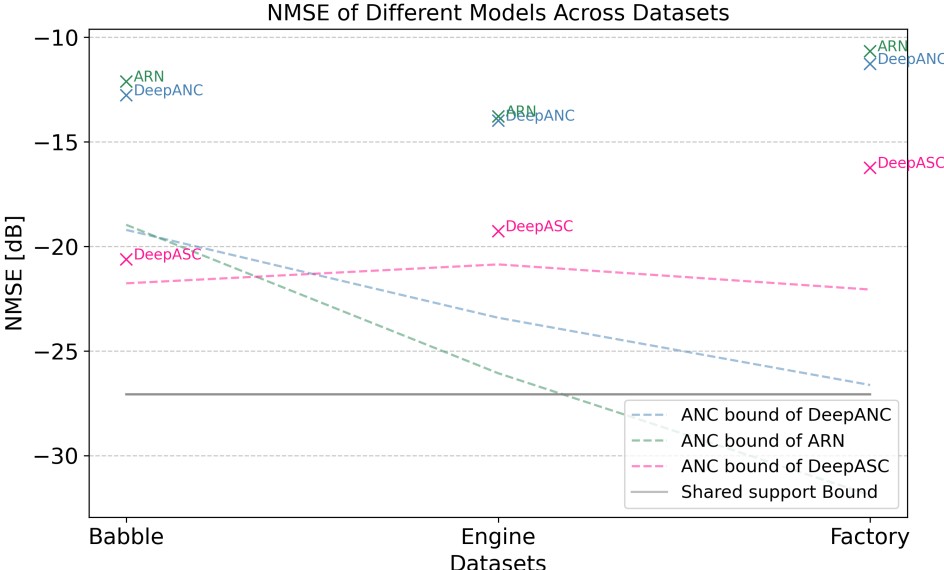

Figure 4: The unified lower bound and NMSE comparison over different noise datasets for $t_{60} = 0.2s$

## B USAGE OF LARGE LANGUAGE MODELS

In this paper Large Language Models (LLM) were used to perform grammatical checks and assist with LaTeX layout (creation and editing of tables and figures).

