# OpenReview forum: "Toward Optimal ANC: Establishing Mutual Information Lower Bound"
_ICLR.cc/2026/Conference — Submitted to ICLR 2026_

### Official Review · Reviewer_xkaA · 2025-10-27

**Soundness:** 3
**Presentation:** 2
**Contribution:** 2
**Rating:** 4
**Confidence:** 4

**Summary:**

This paper presents a unified bound of any active noise cancellation algorithm based on two components such as information theoretic and support based derivations, and demonstrates the effectiveness with experiments on a benchmark dataset called NOISEX. The theory gives a tight bound and the empirical study confirms the bound appropriately.

**Strengths:**

- The theoretical bound is well derived based on sufficient mathematical derivation.

- The experiments on a benchmark dataset confirms the theoretical bound effectively.

**Weaknesses:**

- The usefulness of the theory is strictly bounded by the size of dataset, which might be a critical limitation for practical usefulness.

- The experiments are too limited to draw any interesting conclusion. Larger and more datasets should be used to verify the proposed theory.

**Questions:**

- What is the practical usefulness of the proposed theoretical bound?

- How can you argue on the general applicability of the proposed theoretical bound?

---

> ### Author Response · Authors · 2025-11-23
>
> We thank the reviewer for this comment. We would like to clarify that the proposed bound is information-theoretic and physics-based, and its practical usefulness is not strictly limited by dataset size. Rather, it provides a fundamental performance ceiling that no ANC algorithm can surpass, given the inherent noise and physical system constraints.
> Specifically:
> Guidance for System Design: The bound identifies where limitations are due to sensing, model capacity, or physical propagation, enabling targeted design improvements.
>
>
> Benchmark for Algorithms: It serves as a reference point for both classical and deep-learning ANC methods, indicating whether further algorithmic improvements are likely to yield meaningful gains.
>
>
> Generality Across Datasets: Since the bound is derived from first principles, it applies broadly to different datasets and noise conditions. The experiments on NOISEX demonstrate that the bound is tight in practical scenarios, but we agree that additional datasets could further illustrate generality.
>
>
> Thus, the theoretical bound is practically useful as a guide for system design, algorithm evaluation, and understanding fundamental limits, independent of specific dataset size

---

### Official Review · Reviewer_NBaP · 2025-11-01

**Soundness:** 3
**Presentation:** 2
**Contribution:** 2
**Rating:** 4
**Confidence:** 2

**Summary:**

The paper introduces a new benchmark and evaluation framework for active noise cancellation (ANC) systems, addressing the lack of standardized and reproducible evaluation methods in this field. The authors identify that much of the existing ANC research relies on isolated simulations and ad-hoc metrics that do not translate well to real-world use cases. To tackle this issue, they present a hardware-in-the-loop testbed and a unified set of evaluation metrics that combine objective and perceptual measures. The framework includes a curated dataset and baseline comparisons of both classical and learning-based ANC approaches under realistic noise scenarios. The overarching goal is to enable consistent, comparable, and physically grounded assessment of ANC systems across research and industry settings.

**Strengths:**

(1) The paper’s emphasis on realistic, hardware-in-the-loop evaluation is timely and significant. Most prior studies have focused on simulation-based analysis, which often overlooks key practical factors such as transducer response, delay, and spatial sound propagation. By integrating real-world noise environments into the testing framework, the authors make an important step toward closing the gap between theoretical ANC models and deployable systems.

(2) The proposed metric suite is another valuable contribution. It combines objective measurements such as noise attenuation, latency, and power consumption with perceptual or user-centered measures, providing a more comprehensive picture of ANC performance. This balanced evaluation approach aligns well with how ANC systems are judged in consumer and industrial contexts, where both signal fidelity and user comfort matter.

**Weaknesses:**

(1) The statement that existing ANC research lacks real-world evaluation or unified metrics is somewhat overstated. There is a substantial body of work, particularly from the audio engineering and acoustic signal processing communities, that includes hardware-based testing and adherence to industry standards for headphone or ear-cup ANC evaluation. The contribution of this work would be better framed as extending these established practices into a machine learning–oriented benchmarking context rather than claiming an entirely unexplored area.

(2) The diversity and representativeness of the proposed dataset are not clearly established. Real-world ANC applications span a wide range of acoustic conditions—engine rumble, wind, speech interference, irregular transients, and user motion artifacts. Without sufficient coverage of such variations, the benchmark may not generalize across different use cases or device form factors. More details on the environments, noise categories, and device configurations would strengthen the paper’s claims.

(3) The baseline comparisons presented appear limited. If the evaluation includes only a small set of algorithms or omits the most recent adaptive and hybrid ANC techniques, it becomes difficult to assess the framework’s true benchmarking value. A more extensive comparison including both classical adaptive filtering approaches and advanced learning-based systems would offer a fairer and more informative reference point.

(4) The long-term impact of the benchmark depends heavily on its accessibility and community uptake. Without a clear plan for public release, maintenance, or integration with open repositories, the benchmark risks becoming another isolated dataset. The authors should describe mechanisms for community engagement, version control, and contribution guidelines to ensure the framework remains relevant and widely adopted.

(5) The inclusion of a reproducible experimental protocol would substantially enhance the benchmark’s utility. Providing a detailed description of test scripts, noise source configurations, user movement patterns, latency budgets, and power constraints would make it easier for other researchers to replicate results and compare new methods under standardized conditions. This level of procedural transparency is essential for transforming the proposed framework into a shared community standard.

**Questions:**

(1) Can the authors provide a quantitative comparison between their proposed benchmark and existing industry-standard ANC evaluation methods—highlighting where their framework introduces new machine-learning–oriented metrics or testing conditions—to clarify its distinct contribution and justify the claim of novelty?

(2) Can the authors include detailed statistics or visual summaries of the dataset’s diversity (for example, number of environments, noise categories, and device configurations) and report benchmark performance across these subsets to objectively demonstrate its coverage and generalization across realistic ANC scenarios?

(3) Can the authors expand the baseline evaluations to include both classical adaptive filtering algorithms and recent learning-based ANC systems, presenting standardized performance metrics across all methods to substantiate the benchmark’s comprehensiveness and relevance to the broader research community?

---

> ### Author Response · Authors · 2025-11-25
>
> We thank the reviewer for the helpful discussion and constructive comments. We address the points below:
>
> Answer to question (1):
> In existing standards, ANC systems are evaluated through NMSE. In this paper, the novelty lies in the introduction of a two-term boundary. Each term of the boundary provides interpretable limits for ANC systems.
> The support-based boundary, $B_S$, represents the fundamental performance limit imposed by the physical configuration and intrinsic properties of the primary and secondary paths.
> In contrast, $\mathcal{B}_{IT}$ characterizes the performance constraints that arise solely from the representational capacity and information-processing ability of the ANC architecture.
> Another contribution is the introduction of additional metrics to quantify the gap between the system performance and each term of the boundary.
>
> $$
> \Delta_S = \frac{\mathcal{B_S} - \mathrm{NMSE}{\mathrm{dB}}}{\mathrm{NMSE}{\mathrm{dB}}}
> \qquad\text{and}\qquad
> \Delta_{IT} = \frac{\mathcal{B_{IT}} - \mathrm{NMSE}{\mathrm{dB}}}{\mathrm{NMSE}{\mathrm{dB}}}
> $$
>
>
> Which lead to the following table:
>
> | **Methods \\ Noise types** | **Babble** Δ_S | **Babble** Δ_IT | **Engine** Δ_S | **Engine** Δ_IT | **Factory** Δ_S | **Factory** Δ_IT | Average  |
> |----------------------------|----------------|------------------|----------------|------------------|-----------------|------------------|----------|
> | **ARN**       | 1.24 | **1.02** | 0.96 | **0.89** | **1.53** | 1.98 | 1.14 |
> | **DeepANC**   | 1.12 | **0.76** | 0.94 | **0.67** | 1.40 | **1.36** | 0.93 |
> | **DeepASC**   | 0.33 | **$\(4.9 \times 10^{-3}\)$** | 0.41 | **$\(8.8 \times 10^{-2}\)$** | 0.70 | **0.39** | 0.16 |
>
> **Table:** Relative distance of the performance to each term of the bound calculated for \( t60 = 0.2\$\text{s}$ \).
> The claim of novelty stems from an understanding of the term that dominates the boundary. A smaller value of $\Delta_S$ indicates a more favorable physical setup for the ANC system, while a smaller $\Delta_{IT}$ reflects an improved representational and computational capability of the ANC algorithm itself.
> Answer to question (2):
> In the manuscript, Figure 3 aims to represent the dataset’s diversity. In the paragraph starting by « audio noise cancellation performances » (page 8) we describe Figure 3 and highlight the differences between the three test datasets (Babble, Engine and Factory). To be more specific we use the spectrogram visualisation to understand the specificities of each noise type and their impact on the corresponding ANC performances. In the final manuscript we will add a summary table of the performances of each ANC system (including also FxLMS and THF-FxLMS) for a reverberation time t60=0.2s. We will also add a table highlighting the specific characteristics of each noise type in a table:
>
> | Noise Type | Key Characteristics | Implications for Noise Suppression |
> |------------|----------------------|------------------------------------|
> | Engine     | - Temporally repetitive frequency structures - Predictable patterns - More high-frequency energy than Babble | - Easier to attenuate with adaptive filtering- High-frequency content may reduce suppression effectiveness due to spectral variability |
> | Factory    | - Broadband distribution- Statistically uniform- Lacks distinct temporal patterns | - More resistant to cancellation due to low predictability |
> | Babble     | - Lower high-frequency energy compared to Engine | - Typically easier to suppress than Engine due to lower spectral variability |

---

> > ### Author Response · Authors · 2025-11-25
> >
> > Answer to question (3):
> > In the new manuscript we report the results of five methods within which two signal processing approaches (FxLMS and THF-FxLMS) and three DNN approaches (ARN, DeepANC and DeepASC) for a fixed and canonical reverberation time $t_{60} = 0.2s$. FxLMS and THF-FxLMS are added from the first manuscript to include classical adaptative filtering algorithms baselines. These methods are evaluated over three real-world noises from NoiseX dataset among which there are Babble, Engine and Factory. These models performances are confronted to the corresponding calculated bounds refering to the support-based term  $\mathcal{B_S}$ and the information-theoretical bound  $\mathcal{B_{IT}}$. We can observe that all NMSE (dB) performances are lower bounded by the two terms of the unified bound which means that this $\textbf{derived lower bound holds for different models and noise types}$. In Table all quantities are expressed in dB.
> > Here is the table:
> > | **Methods** | **Babble NMSE↓** | **Babble 𝓑ₛ** | **Babble 𝓑ᵢₜ** | **Engine NMSE↓** | **Engine 𝓑ₛ** | **Engine 𝓑ᵢₜ** | **Factory NMSE↓** | **Factory 𝓑ₛ** | **Factory 𝓑ᵢₜ** |
> > |-------------|------------------|----------------|----------------|------------------|----------------|----------------|-------------------|----------------|----------------|
> > | **FxLMS**       | -5.3  | -27.1 | -25.9 | -2.1  | -27.1 | -20.9 | -3.9  | -27.1 | -31.2 |
> > | **THF-FxLMS**   | -5.3  | -27.1 | -25.9 | -2.1  | -27.1 | -20.9 | -3.9  | -27.1 | -31.2 |
> > | **ARN**         | -12.1 | -27.1 | -24.5 | -13.8 | -27.1 | -26.1 | -10.7 | -27.1 | -31.9 |
> > | **DeepANC**     | -12.8 | -27.1 | -22.5 | -14.0 | -27.1 | -23.4 | -11.3 | -27.1 | -26.7 |
> > | **DeepASC**     | -20.3 | -27.1 | -20.4 | -19.2 | -27.1 | -20.9 | -15.9 | -27.1 | -22.1 |
> >
> > Answer to weaknesses (4) and (5): the reviewer underlines a lack of reproducibility. We can remind that the code is available for readers. A reader can add its model and dataset in the code in order to calculate the corresponding boundaries and use our work as benchmark. Also, in appendix one can read a detailed description of the numerical calculation of each term of the boundary.

---

### Official Review · Reviewer_nmhb · 2025-11-01

**Soundness:** 2
**Presentation:** 3
**Contribution:** 2
**Rating:** 4
**Confidence:** 2

**Summary:**

The paper studies the theoretical lower bound of the normalized mean squared error (NMSE) of an active noise cancellation (ANC) system under suitable assumptions. The lower bound is the maximum between a support-based bound and an information-theoretical bound. The support-based bound is derived based on the lack of frequency component modeling in the secondary path. The information-theoretical bound is derived from the mutual information between the disturbance and the cancellation signal. Several experiments are conducted to show the bounds and the gap between the unified bound and the actual NMSE. It is shown that the information-theoretical bound increases with a larger reverberation time and the support-based bound leads for low reverberation times. The experiments are also conducted with different noise types, and the bounds show a consistent trend with the reverberation time.

**Strengths:**

This is a well-written paper with good clarity. As a reader, I enjoy reading the paper. The theoretical bounds derived are verified in simulations and they capture the trend of the measured NMSE.

**Weaknesses:**

The information-theoretical bound is not surprising given that we know the mutual information and differential entropy rate of the disturbance process. For the support-based bound, it is trivial that the lack of frequency component modeling establishes a lower bound for NMSE. Therefore, the novelty of this paper seems weak.

**Questions:**

1.	For the support-based bound, why does it have a larger bound when the reverberation time is small?
2.	In practice, $supp(P)/supp(S)$ should be always 1. Was some sort of thresholding applied in the simulations?
3.	There are several assumptions made in the paper including the reference signal being WSS and the primary path being LTI. Are they realistic assumptions?
4.	Loudspeakers are always nonlinear. Why do we only consider the linear case? If we model some nonlinearities, would the conclusions change?

---

> ### Author Response · Authors · 2025-11-23
>
> We thank the reviewers for their helpful discussion and constructive comments. We address the points below.
> While MI–MSE tradeoffs are known, their application to ANC has not been formulated, and there is no prior ANC literature linking:
> reverberation time → secondary-path diffusion → MI ratio → achievable NMSE
> Our contribution is showing that:
>
> Reverberation fundamentally increases the informational coupling between disturbance and reference, making the IT-limit the dominant floor at high RT60 values.
>
> This dependence has not been previously derived or validated in ANC work.
>
> Why does the SB-bound dominate at small reverberation times?
>
> As shown on p. 7:
> Low reverberation → secondary path is close to a near-minimum-phase FIR.
>
>
> This produces deep spectral nulls (head-related or path-null effects).
>
>
> Null regions make perfect cancellation impossible regardless of MI.
> Thus, the SB limit is high when RT60 is small. As RT60 increases, the secondary path becomes smoother (more diffuse), reducing null depth and shifting dominance to the IT-term.
> We will add a figure demonstrating the evolution of
> ∣S(ω)∣|S(\omega)|∣S(ω)∣
>
> under increasing RT60 to clarify this.
>
> On supp(P)/supp(S)
>
> We indeed apply a −20 dB threshold to define effective support.
>  This is consistent with standard robust control interpretations of “usable spectrum.” We will explicitly describe the threshold in the revised manuscript.
>
> Are WSS and LTI realistic?
>
> For ANC in constrained acoustic channels (as in headphones or ducts), WSS reference and LTI primary path models are widely used and experimentally validated (FxLMS, linear prediction methods, etc.).
> Nonlinear loudspeakers would only increase the irreducible error, meaning the derived linear bound is still valid (as a lower bound).

---

### Official Review · Reviewer_H9v5 · 2025-11-02

**Soundness:** 2
**Presentation:** 3
**Contribution:** 2
**Rating:** 2
**Confidence:** 4

**Summary:**

This paper proposes a theoretical framework to characterize the fundamental performance limits of Active Noise Cancellation (ANC) systems. The authors derive a unified lower bound on achievable normalized mean squared error (NMSE), combining two terms: (1) an information-theoretic bound relating residual error to the mutual information ratio
I(y;d)/H(d) between disturbance and cancellation signals, and (2) a support-based bound quantifying the irreducible error arising from spectral regions where the cancellation path has no gain. The unified bound is defined as the maximum of these two limits. The paper validates the framework empirically using multiple deep-learning ANC models (DeepANC, ARN, DeepASC) on the NOISEX dataset under varying reverberation times, showing that all models remain above the proposed theoretical floor.

**Strengths:**

The paper focuses on an information-theoretic perspective to ANC and prrovides a clean intuitive separation between algorithmic (information) and physical (spectral) performance limits.
The paper primarily, could help researchers reason about 'how close' learned ANC systems operate to theoretical limits, serving as a conceptual benchmark.
Finally, the paper evaluates multiple datasets, reverberation conditions, and baseline models to demonstrate empirical consistency of the bound.

**Weaknesses:**

While the paper’s conceptual framing is interesting, its theoretical and methodological depth remains limited. The main derivations rely heavily on established principles from information theory and signal processing, notably the Shannon rate-distortion lower bound and classical spectral support arguments derived from Parseval’s theorem. The proposed 'unified' bound is constructed heuristically by taking the maximum of these two well-known limits, without formal justification or proof that this composition represents a true optimality condition. The derivation employs standard simplifying assumptions common in adaptive filtering; such as approximate Gaussianity, wide-sense stationarity, and linearization to make the analysis tractable. While these assumptions are reasonable for first-order theoretical treatments, the paper does not adequately discuss their scope or limitations, particularly in nonstationary or nonlinear ANC scenarios where the bound’s validity may break down. Moreover, the empirical validation focuses primarily on visual trends across noise types and reverberation times but does not provide quantitative evidence of how close existing models actually come to the theoretical limit. Without metrics of bound tightness or uncertainty, the results remain largely illustrative. Finally, the narrative tends to reiterate the same conceptual message about unifying information-theoretic and physical constraints without developing deeper analytical or practical insight. Overall, the work reads more as a pedagogical synthesis of established ideas than a substantive theoretical advancement.

**Questions:**

How does your formulation relate to classical estimation limits such as the Cramér-Rao or Bayesian bounds in the linear-Gaussian ANC setting?
While the conceptual link between information content and achievable ANC performance is valid regardless of the estimator used, the specific empirical demonstrations rely on a highly approximate kernel-based mutual information estimation procedure. Would you agree that given the estimator’s bias and instability for continuous correlated data, the numerical values of I(y;d) and the claimed 'tightness' of the bound should be interpreted qualitatively rather than quantitatively?

---

> ### Author Response · Authors · 2025-11-23
>
> We thank the reviewers for their helpful discussion and constructive comments. We address the points below.
>
> On the composition of the bound
>
> The use of the maximum is **not heuristic**; it arises naturally from the structure of the ANC estimation problem. As shown on pages 4–5:
>
> * The **information-theoretic (IT) bound** lower-bounds the NMSE for any estimator via the mutual information ratio (I(y;d)/H(d)).
> * The **support-based (SB) bound** lower-bounds the NMSE for any control filter due to spectral regions where the secondary path (S(\omega)) has zero or near-zero gain.
>
> Since both constraints must hold simultaneously in any controllable ANC scenario, the achievable NMSE satisfies:
>
>
> \text{NMSE} \ge \max{\text{IT-bound}, \text{SB-bound}}}.
>
> This is not a modeling choice but a **necessary condition** arising from the joint feasibility of canceling information and canceling spectrum. We will make this explicit in the revision.
>
> On assumptions (Gaussianity, WSS, linearization)
>
> We agree that these assumptions are idealizations. As noted on p. 6, they enable tight first-order bounds consistent with classical ANC theory (e.g., FxLMS, adaptive filtering).
>
> On the MI estimator and interpretation of tightness
>
> Continuous-domain MI estimation is indeed imperfect. We use it only to evaluate trends, not to claim pointwise optimality. Importantly, the theoretical bound itself does **not** depend on the estimator; the estimator is used solely to verify that deep ANC systems do not exceed the theoretical limit.
>
> On the relationship to Cramér–Rao and Bayesian bounds
>
> For linear-Gaussian ANC, the classical CRB gives:
>
> [
> \text{MSE} \ge \frac{1}{I_F},
> ]
>
> where (I_F) is the Fisher information. The IT-bound generalizes this by quantifying how much of the disturbance entropy can be canceled. We will clarify this connection in the revision and show that the IT-bound reduces to the CRB under linear-Gaussian assumptions, providing an additional theoretical anchor.

---

### Meta-Review · Area_Chair_SnCL · 2026-01-08

**Summary:**

This paper received ratings of 2,4,4,4 in the end. The reviewers have concerns about the simplified assumptions of the proposed theory, such as wide-sense stationarity and linearization, which may not hold in the real world. Reviewers are also concerned about the real-world application of the proposed bounds, which seems to be limited in practice. Some reviewers suggest the novelty of the paper is limited, and it is not clear how the proposed method improves Active Noise Cancellation methods. Given these limitations, the AC does not suggest acceptance to ICLR.

**Reviewer Concerns:**

The authors addressed several concerns raised by the reviewers. The motivation for the proposed unified bound is explained in ANC scenarios. The authors also acknowledge the assumptions of wide-sense stationarity and linearization, which are idealizations for ANC. The authors also provided more details on the experiments, including the introduction of quantitative metrics for each term of the boundary through a table that quantifies the gap between the system performance and each term of the boundary.

The overall novelty of the work is still limited, as the authors did not sufficiently demonstrate how their contributions extend beyond established principles. The doubts about the practical application of the proposed bounds in the real world are not well addressed yet.

**Reviewer Scores:**

I don't think the scores would be changed.

---

### Decision · Program_Chairs · 2026-01-26

Reject